# Residual Vision Transformer and Adaptive Fusion Autoencoders for Monocular Depth Estimation

**DOI:** 10.3390/s25010080

**Published:** 2024-12-26

**Authors:** Wei-Jong Yang, Chih-Chen Wu, Jar-Ferr Yang

**Affiliations:** 1Department of Artificial Intelligence and Computer Engineering, National Chin-Yi University of Technology, Taichung 411, Taiwan; wjyang@ncut.edu.tw; 2Institute of Computer and Communication Engineering, Department of Electrical Engineering, National Cheng Kung University, Tainan 701, Taiwan; jesse90302@gmail.com

**Keywords:** monocular depth estimation, convolutional neural networks, residual vision transformer, adaptive fusion, autoencoder

## Abstract

Precision depth estimation plays a key role in many applications, including 3D scene reconstruction, virtual reality, autonomous driving and human–computer interaction. Through recent advancements in deep learning technologies, monocular depth estimation, with its simplicity, has surpassed the traditional stereo camera systems, bringing new possibilities in 3D sensing. In this paper, by using a single camera, we propose an end-to-end supervised monocular depth estimation autoencoder, which contains an encoder with a structure with a mixed convolution neural network and vision transformers and an effective adaptive fusion decoder to obtain high-precision depth maps. In the encoder, we construct a multi-scale feature extractor by mixing residual configurations of vision transformers to enhance both local and global information. In the adaptive fusion decoder, we introduce adaptive fusion modules to effectively merge the features of the encoder and the decoder together. Lastly, the model is trained using a loss function that aligns with human perception to enable it to focus on the depth values of foreground objects. The experimental results demonstrate the effective prediction of the depth map from a single-view color image by the proposed autoencoder, which increases the first accuracy rate about 28% and reduces the root mean square error about 27% compared to an existing method in the NYU dataset.

## 1. Introduction

The purpose of depth estimation is to accurately predict the distance between objects and the camera. Depth information finds wide applications across various fields, including household robots [1], autonomous driving [2], 3D movie production [3], etc. Depth information could also be the input data for other computer vision tasks such as face recognition [4], object detection [5] and semantic segmentation [6]. A high-quality depth map is mostly characterized by accurate depth values specified along well-defined object boundaries.

Depth estimation by stereo matching neural networks was first initiated by comparing image patches [7]. To further improve the depth quality, the two-stage network with cascade residual learning [8], the pyramid network [9] and the network with semi-global and mutual information [10] were proposed. Stereo matching neural networks, however, need multiple cameras for depth prediction. If we try to use a single-view image to predict depth, the estimation process becomes a challenging task due to its ill-posed condition. When humans look at a picture and try to understand the spatial relationship of the objects from it, we mostly consider both local cues and global context. Local cues refer to details such as the texture appearance and the perspective of objects, the relative sizes, etc. On the other hand, global context, referring to occlusion issues and global spatial relationships, could be exhibited from the layout of the scene. By assessing these factors, humans and the monocular depth estimation neural networks can make good sense of the geometric configuration from a single image.

For deep learning networks, mainly as the feature extractors, we could employ a series of convolutional neural and down-sampling blocks to gradually extract the detailed and global features layer-by-layer. For instance, VGG [11] achieves this by applying multiple 3 × 3 convolution layers and pooling operations to encode the image into latent features. For better convergence, ResNet [12] utilizes residual blocks with skip connections to learn residual information and extract image features. For both VGG and ResNet, the features in the shallow layers possess more detailed information while those in the deeper layers hold more global information. In recent years, vision transformers [13] have gained a lot of attention because they are with rich global information and can achieve a good performance in computer vision tasks. Many researchers attribute this success to the self-attention mechanism [14], which enables the input features to capture abundant global information and significantly expand their receptive fields. However, the amounts of parameters and calculations of the vision transformers are very large. In addition to the final extracted features, we must efficiently utilize the detailed features from lower layers by the co-called skip connections [15]. How to perfectly fuse the decoded feature and the skip-connected encoder feature is also crucial in the design of the decoder.

## 2. Related Work

Monocular depth estimation (MDE) [16,17], which is accomplished by using a single color image, can significantly minimize the requirement for multiple cameras and greatly reduce hardware resources. Since monocular depth estimation methods only take a single color image as the input, it is more difficult to estimate the precision depth map than with the stereo matching approaches. Heavily depending on the image to ground truth depth mapping regression, monocular depth estimation methods with deep learning structures can be divided into supervised learning [18], unsupervised learning [19] and semi-supervised learning approaches [20]. Supervised networks apply ground truth depth maps to train a neural network as a regression model. Eigen et al. [18] were the pioneers to approach monocular depth estimation using a deep learning method, where the CNN-based network comprises two stacked deep networks, a coarse depth network and a refinement network. As regards unsupervised monocular depth estimation, Godard et al. [19] introduced a network system that takes a single-view image as the input to generate depth map without ground truth depth. During the training period, however, it needs both the left and right views. The input left and right view images with the estimated depth maps by the MDE networks are warped to the other synthesized view images. The reconstruction loss subsequently utilizes the closeness of the synthesized and the input images to facilitate unsupervised learning. The semi-supervised learning approach [20] simultaneously apply supervised and unsupervised loss terms; however, the possible synthesizing of paired images should be performed. Considering consecutive multiple single-view frames, Yang et al. [21] suggested video-based depth estimation autoencoder to further improve the depth performances.

### 2.1. Vision Transformer

Dosovitskiy et al. [13] are the pioneers of developing a structure related to a transformer in an image classification task. The vision transformer (ViT) [13], which is a new type of neural network for computer vision, extends the success of transformers originally developed for natural language processing [14]. To extract more global features, the ViT and its variations have gained significant attention and achieved state-of-the-art results in various computer vision tasks such as image classification [22], semantic segmentation [23] and depth estimation [24] but with higher computation. Unlike traditional convolutional neural networks (CNNs) that rely on spatial convolutions and pooling layers, the vision transformers utilize the multi-head self-attention mechanism to calculate global dependencies and long-range relationships within an image.

In vision transformers, the input image is segmented into patches, which are then flattened to vectors. Linear projection or 1 × 1 convolution is used for adjusting the length of the flattened vectors as “patch embeddings” or “tokens”. These tokens are then passed through the transformer blocks to capture global information. The basic architecture of a vision transformer block is shown in Figure 1, where the ViT is used to detect the class (bird, car, ball, or ….) of the image. The vision transformer consists of layer normalization (LN), multi-head self-attention (MSA), channel MLP and the residual connections.

Many studies have proposed using a vision transformer to increase global information [25,26,27] for monocular depth estimation. With a Transbins module, Depthformer [26] employs the attention-based architecture to attain the advantage of a global reception field. To explore local details and global dependency, PCTNet [27] adopts CNN and vision transformer branches by using bidirectional feature and cross-feature multi-scale fusion modules to obtain good results for single structured light image depth estimation. GlobalFuse-Depth [28] also suggests CNN and vision transformer branches to capture the features of paired images and fuse them together to achieve better depth estimation. The network needs paired, i.e., night and daytime, images with a pretrained CycleGAN. It is noted that PCTNet uses self-supervised training and tests with structured light databases. Generally, the vision transformers will introduce huge computation in their networks.

### 2.2. Atrous Spatial Pyramid Pooling

To expand the receptive field for convolutional neural networks, a common solution is to increase the kernel size of standard convolutions. However, the computation will become larger while using a bigger convolutional kernel size. Dilated convolution is similar to standard convolution by introducing gaps between each kernel pixel based on the specific dilation rate. The dilated convolution allows the kernel receptive field to be expanded without increasing computations. For instance, in a standard 3 × 3 convolution with dilation rate of two, its receptive field is expanded to achieve the same receptive field as a standard 5 × 5 standard convolution kernel. The 3 × 3 convolution with a dilation rate of two utilizes only nine kernel parameters.

Chen et al. [29] introduced the Atrous spatial pyramid pooling (ASPP) module, as shown in Figure 2. The ASPP module learns a comprehensive feature by combining the features obtained from a pooling layer and multiple convolution layers with different dilation rates. The BTS network [30] is composed of a dense feature extractor, ASPP as the contextual information extractor, local planar guidance layers and their dense connection for depth estimation. The ASPP captures the large-scale variations in features by applying sparse convolutions with various dilation rates. The BTS network presents a supervised monocular depth estimation network to achieve state-of-the-art results.

### 2.3. Selective Feature Fusion

To effectively fuse the skip connection features, GLPDepth [17] utilizes a selective feature fusion (SFF) module to achieve high-quality depth, as shown in Figure 3. Instead of element-wise summation of skip-connected encoder and decoder features, the SFF module offers improved fusion capabilities. It is noted that the skip connection feature *F_c_* and the decoder feature *F_d_* have the size of *C* × *H* × *W*. They are first concatenated along the channel dimension and then passed through two layers of 3 × 3 convolution, batch normalization, and ReLU and finally through a 3 × 3 convolution layer to reduce the number of channels to two, and undergo sigmoid activation function to obtain two separate attention maps, *A_c_* and *A_d_*. By performing element-wise multiplications of *F_c_* to *A_c_* and *F_d_* to *A_d_*, these weighted features after the element-wise summation construct the final fused feature *F_f_*.

## 3. Proposed Methods

For monocular depth estimation, effectively combining the local information and the global features is an important challenge for the networks to estimate the depth map with exceptional quality. As shown in Figure 4, the basic framework of the proposed residual vision transformer and adaptive fusion (RVTAF) depth estimation network consists of a CNN-ViT encoder and an adaptive fusion decoder. The CNN-ViT encoder is further composed of a CNN feature extractor mixed with several ViT modules in residual configurations to extract local and global features, while the multiple-level features are skip-connected to fuse the feature of the decoder to achieve high-quality depth estimation. In the RVTAF depth estimation network, we need to identify a better residual configuration of vision transformers to successfully expand the receptive field of the bottleneck feature. We also need to design an effective adaptive fusion module to further enhance the precision of the estimation. A detailed explanation of the CNN-ViT encoder and adaptive fusion decoder is present in the following two subsections.

### 3.1. CNN¬-ViT Encoder

The detailed structure of the final CNN_ViT encoder, which is shown in Figure 5, mainly contains subsampled residual blocks (SRBs) and residual blocks (RBs) to extract 3 intermediate features, *F*_1_, *F*_2_, *F*_3_, and the final bottleneck feature *F*_o_ with a size of 512 × H/16 × W/16. Through experiments, the proposed CNN-ViT encoder incorporates vision transformers (ViTs), which are marked in yellow color, and will be further discussed in the subsequent sections. Inspired by ResNet50 [12], the backbone is constructed by two different building blocks to become the CNN feature extractor. In a formulation, the CNN-ViT encoder with the input of the image, *I*, and outputs of *F*_1_, *F*_2_, *F*_3_ can be expressed as follows:{*F*_1_, *F*_2_, *F*_2_, *F*_o_} = CNN-ViT(*I*)(1)
where the front CNN part of the encoder can be mainly expressed by several subsampled residual blocks (SRBs) and residual blocks (RBs). Thus, the computation of intermediate skip features can be further given as follows:*F*_1_ = RB2(SRB(*I*)), *F*_2_ = RB3(SRB(*F*_1_)), *F*_3_ = RB5(SRB(*F*_2_))(2)
and the final output with ViT stages is given as follows:*F*_0_ = ViT3(SRB(*F*_4_)) with *F*_4_ = RB2(SRB(*F*_3_)) + ViT2(SRB(*F*_3_))(3)
where the numbers behind the RB and ViT functions denote the repeated number of the modules. The detailed structures of the subsampled residual block (SRB) and residual block (RB), and the vision transformer (ViT) are shown in the following three subsections.

#### 3.1.1. Subsampled Residual Block

As shown in Figure 6, the subsampled residual block (SRB) has two branches, where the lower branch first projects the input feature to a lower dimension space by 1 × 1 convolution (Conv1) to reduce the number of channels by half. Then, the spatial information is further down-sampled through a 3 × 3 convolution with stride 2, i.e., Conv3_s2, and finally the features are projected to twice dimension of the input feature with 1 × 1 convolution. The upper branch first uses a max-pooling operation to downsize the spatial information by half, followed by a 1 × 1 convolution (Conv1) to double the channel number. Finally, the features from these two branches are combined through element-wise summation to obtain the output SRB feature. The height and width of the input feature (*C* × *H* × *W*) are reduced to half while the channel number is increased twice for the output feature (2*C* × *H*/2 × *W*/2). In formulation expression, the SBR module with the input feature *f_i_* and the output feature *f_i_*_+1_ can be expressed as follows:*f_i_*_+1_ = SRB(*F_i_*)) = ReLU{SRBU(*f_i_*) + SRBB(*f_i_*)}(4)
where the upper and bottom parts of SRB module are, respectively given as follows:SRBU(*f_i_*) = Conv1(Maxpool(*f_i_*))(5)
and
SRBB(*f_i_*) = BN(Conv1(ReLU(BN(Conv3_s2(Relu(BN(Conv1(*f_i_*))))))(6)

#### 3.1.2. Residual Block

As shown in Figure 7, the residual block (RB) employs a 1 × 1 convolution to reduce the dimension of the input feature by half. Once the feature is projected into the low-dimensional space, we utilize a 3 × 3 convolution with a stride of 1, i.e., Conv3, to capture spatial information. Following that, a 1 × 1 convolution (Conv1) is used to adjust the dimension to match that of the input feature. Finally, we combine the learned feature with the input feature through element-wise summation. The output feature of the residual block (RB) maintains the same feature size as the input feature. In formulation expression, the RB module with input feature *f_i_* and output feature *f_i_*_+1_ can be expressed as follows:*f_i_*_+1_ = *f_i_* + BN(Conv1(ReLU(BN(Conv3(ReLU(BN(Conv1(*f_i_*))))))))(7)

#### 3.1.3. Vision Transformers in CNN-ViT

The flow chart of the realized vision transformer is shown in Figure 8. The input feature map, *f,* is segmented into patches with a size of *p* × *p*, where we set *p* = 5, to obtain *N* = *HW*/*p* patches. These patches are flattened into vectors, followed by a 1 × 1 convolution to adjust the length of vectors from *C* to *C_vit_*. We call the *i*th adjusted vector “the *i*th token”, *t_i_*, which has the size of *C_vit_* × 1 × 1. After the preparation of the inputs for vision transformer, these tokens are sent into the vision transformer, which is shown in Figure 1b, to learn global information. The vision transformer first maps the *i*th token *t_i_* into query (*q_i_*), key (*k_i_*) and values (*v_i_*) as follows
(8)$$q_{i}=W_{q}t_{i},\;k_{i}=W_{k}t_{i},\;v_{i}=W_{v}t_{i}$$
where *W_q_*, *W_k_* and *W_v_*, respectively, denote the linear transformations for queries, keys and values. Let the initial patch matrix as *z*_0_ = [*t*_1_, *t*_2_, …., *t_N_*]; the *j*th iteration of multi-head self-attention (MSA) is shown as follows:(9)$$\mathrm{MSA}(z_{j})=\mathrm{Attention}\left(Q,K,V\right)=\mathrm{softmax}\left(\frac{QK^{T}}{\sqrt{M_{vit}}}\right)V$$
where *Q* = {*q*_1_, *q*_2_, …, *q_N_*}, *K* = {*k*_1_, *k*_2_, …, *k_N_*} and *V* = {*v*_1_, *v*_2_, …, *v_N_*}. As shown in Figure 6, the *n*-iteration vision transformer can be expressed as follows:(10)$${z^{\prime }}_{j}=\mathrm{MSA}(\mathrm{LN}(z_{j-1}))+z_{j-1},\mathrm{for}\;j=1,2,\ldots ,n$$
(11)$$z_{j}=\mathrm{MLP}(\mathrm{LN}({z^{\prime }}_{j}))+{z^{\prime }}_{j},\mathrm{for}\;j=1,2,\ldots ,n$$

In (2), the ViT2 and ViT3 functions perform *n* = 2 and *n* = 3 iterations of the vision transformer process defined in (9)–(11), respectively.

After the vision transformer, the patch embeddings, which have the size of *M_vit_* × 1 × 1, learn a lot of global information. We deploy a 1 × 1 convolution to adjust the lengths of the learned patch embeddings from *C_vit_* to *p* × *p* × *C*. Then, as the flattened procedure, we reverse the process and restore the learned patch embeddings back to their original *C × H × W* size.

There are many ways to insert the vision transformers (ViTs) into the CNN encoder. Initially, we attempted to insert vision transformers into the feature extraction pipeline in sequential manners; however, we found that the series configurations cannot improve the quality of depth estimation. To mitigate the impact of computation, we should not put the vision transformers in the shallow layers. Thus, we proposed a general residual layout of the vision transformers into the CNN feature extractor as shown in Figure 9, where we mark five positions (Position 1, Position 2, …., Position 5) for adding (*n*_1_, *n*_2_…., *n*_5_) ViTs, respectively. With the limitation of 5 ViTs, i.e., *n*_1_ + *n*_2_ + *n*_3_ + *n*_4_ + *n*_5_ = 5 for the consideration of reasonable computation complexity, we determine a better configuration of these five residual ViTs in the CNN network through the experiments detailed in Section 4.

### 3.2. Adaptive Fusion Decoder

To achieve a good decoder, we believe that feature fusion of skip connections will be the crucial design to achieve an effective autoencoder. The decoder layer could refer to the detailed information of the encoder, which progressively extracts layer features with a global scope. Consequently, effectively integrating the skip connection feature extracted from the encoder with the decoded feature returned by the decoder becomes an indispensable concern. The structure of the adaptive fusion decoder as shown in Figure 10 is composed of up-conversion (Upconv) blocks, fusion modules (FMs) and a Deep ASSP module. In formulation expressions, the adaptive fusion decoder, whose inputs are the connected features *F*_1_, *F*_2_, *F*_2_ and the encoder output features, *F_o_*, finally estimates the estimation depth as follows:(12)$$\overset{⌢}{d}=\mathrm{AFdecorder}(F_{1},F_{2},F_{3},F_{o})$$

To fuse the connected features one-by-one, the above AFdecoder process can be further decomposed into the following processes as follows:(13)$$F_{3}^{d}=\mathrm{FM}(\mathrm{Upconv}(F_{o}),F_{3}))$$
(14)$$F_{2}^{d}=\mathrm{FM}(\mathrm{Upconv}(\mathrm{DeepASPP}(F_{3}^{d}),F_{2}))$$
(15)$$\overset{⌢}{d}=\mathrm{Upconv}(\mathrm{FM}(\mathrm{Upconv}(F_{2}^{d}),F_{1}))$$

The detailed descriptions of fusion modules (FMs), up-conversion (Upconv) block, and Deep ASSP module will be explained in the following subsections.

#### 3.2.1. Fusion Modules

There are many ways to fuse two features together. To improve the SFF module [16], we suggest three variants of the fusion module (FM), namely, separate enhancement addition fusion module (SEAFM), separate enhancement concatenation fusion module (SECFM) and adaptive fusion module (AFM). We believe that the connected feature *F_c_* and the decoded feature *F_d_* have their distinct feature characteristics, which implies that their attention maps *A_c_* and *A_d_* cannot be generated using the same set of weights and need to be extracted separately. The fusion module (FM) performs the fusion process as follows:(16)$$F_{f}=\mathrm{FM}(F_{c},F_{d})=F_{c}⊗A_{c}+F_{d}⊗A_{d}$$
where ⊗ denotes the element-by-element multiply operator. To find the attention maps, the detailed explanations of these three fusion modules are presented as follows.

A.Separate Enhancement Addition Fusion Module

As shown in Figure 11, the separate enhancement addition fusion module (SEAFM) independently enhances the two input features. Comparing to Figure 3, the skip connection feature branch and the decoded feature branch undergo two sequential 3 × 3 convolution-batch normalization-ReLU layers to reduce the number of channels to one-fourth of the original.

B.Separate Enhancement Concatenation Fusion Module

As shown in Figure 12, the separate enhancement concatenation fusion module (SECFM) is a modified version of the SEAFM by replacing the addition operation of the SEAFM with the concatenation operation of two weighted features, which are then further processed by a 3 × 3 convolution, batch normalization and ReLU operations. As expected, the element-wise summation in the SEAFM is slightly more efficient and requires fewer parameters than the concatenation in the SECFM.

C.Adaptive Fusion Module

As shown in Figure 13, the adaptive fusion module (AFM) first concatenates the skip connection feature and the decoded feature before being split into two branches for attention map generations. This approach ensures that both branches access the information from two features, thereby enhancing the generation process with more comprehensive and integrated information. By incorporating this strategy, the attention maps *A_c_* and *A_d_* can effectively prioritize crucial information from both features, leading to a better improved performance. It is noted that the SEAFM, which independently generates two attention maps, lacks the knowledge of the information present in the input features. The adaptive fusion module with a better reference of them through the first concatenation can adaptively generate better attention maps.

#### 3.2.2. Up-Convolution Module

We deploy up-convolution (Upconv) blocks to increase the width and height of the decoded feature and fused features while reducing the number of channels. This step ensures that the feature not only matches the input size of the subsequent AFM but also enhances the precision of spatial information in the up-sampled feature. The architecture of the up-convolution block is shown in Figure 14. The input feature of the up-convolution block will first up-sample to double its width and height, and then pass through a layer of 3 × 3 convolution-batch normalization-ReLU for enhancing the spatial information. In (13–15), the Upconv function is given as follows:(17)$$F^{\prime }=\mathrm{Upconv}(F)=\mathrm{ReLU}(\mathrm{BN}(\mathrm{Conv}(\mathrm{Upsample}(F))))$$

#### 3.2.3. Deep ASPP Module

When the dilation rate exceeds the width and height of the feature, dilated convolution behaves similarly to a 1 × 1 convolution. Consequently, the output of certain branches in the ASPP module cannot extend the receptive field. To improve Atrous spatial pyramid pooling (ASPP) [29,30], we deploy the Deep ASPP [31], which was used for the segmentation task, to help the model to expand its receptive field of the feature. Unlike the original ASPP, the Deep ASPP possesses a wider receptive field, and prevents the degradation of Atrous convolution kernels with high dilated rates into 1 × 1 convolution. We use the Deep ASPP module after the first fusion module (FM), as shown in Figure 10. The architecture of the Deep ASPP module is shown in Figure 15; thus, the DeepASPP function in (14), based on Figure 15, can be expressed as follows:(18)$$F_{3}^{\prime d}=\mathrm{DeepASPP}(F_{3}^{d})=\mathrm{Conv}(\mathrm{ELU}(\mathrm{Concat}[F_{3}^{d},F_{3,3}^{d},F_{3,6}^{d},F_{3,12}^{d},F_{3,18}^{d},F_{3,24}^{d}]))$$
where
(19)$$F_{3,3}^{d}=\mathrm{ACB\_}3(\mathrm{BN}(\mathrm{ELU}(\mathrm{Conv}(F_{3}^{d})))$$


(20)
$$F_{3,6}^{d}=\mathrm{ACB\_}6(\mathrm{Concat}[F_{3}^{d},F_{3,3}^{d}])$$



(21)
$$F_{3,12}^{d}=\mathrm{ACB\_}12(\mathrm{Concat}[F_{3,3}^{d},F_{3,6}^{d}])$$



(22)
$$F_{3,18}^{d}=\mathrm{ACB\_}12(\mathrm{Concat}[F_{3,3}^{d},F_{3,6}^{d},F_{3,12}^{d}])$$



(23)
$$F_{3,24}^{d}=\mathrm{ACB\_}24(\mathrm{Concat}[F_{3,3}^{d},F_{3,6}^{d},F_{3,12}^{d},F_{3,18}^{d}])$$


**Figure 15 sensors-25-00080-f015:**
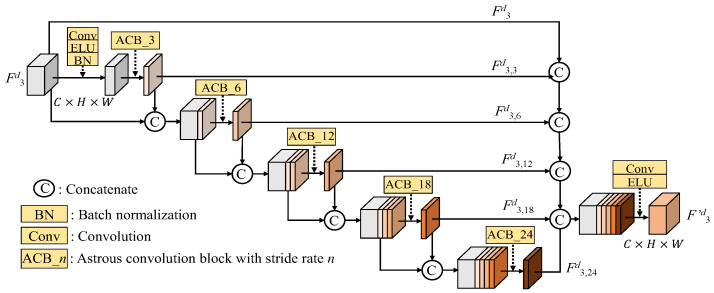
The architecture of the Deep ASPP.

### 3.3. Training Loss Function

In order to calculate the distance between the predicted depth map $\hat{D}$ and the ground truth depth map *D*, we use a scale-invariant log loss [18] to train the proposed network. The training loss function is given as follows:(24)$$L\left(d_{i},{\hat{d}}_{i}\right)=\frac{1}{n}{\sum_{i}{y_{i}}^{2}-\frac{\alpha }{n^{2}}\left(\sum_{i}y_{i}\right)}^{2}$$
with *y_i_* measuring the loss of the *i*th log depth as follows:(25)$$y_{i}=\log d_{i}-\log{\hat{d}}_{i}$$
where $d_{i}$ and ${\hat{d}}_{i}$ represent the ground truth depth and predicted depth of the *i*th pixel, respectively. In (1), the loss function is calculated with square-mean minus mean-square, known as the variance when α = 1. When α = 0, the loss function becomes an L2 loss. Here, we set α = 0.5 to train our network, as suggested in [18].

During the network training process, we normalize the ground truth depth values to a range between 0 and 1. This normalization allows the network to predict depth values within the same range during regression. By applying the logarithm function as shown in Figure 15 to the difference between two values ranging from 0 to 1, the error is effectively amplified, especially for low values. This amplification has a stronger impact on smaller ground truth depth values. As a result, our predicted depth map prioritizes the accurate prediction of foreground depth values. This amplification helps in improving the accuracy of our predicted depth values. It is important to highlight that during the training process with the KITTI dataset, we exclusively consider pixels where $d_{i}>0$ and ${\hat{d}}_{i}>0$ for calculating the loss function. This approach is adopted because logarithms of depth values cannot be defined when the depth is 0. By focusing on non-zero depth values, we prevent difficulties that may arise during training.

## 4. Experimental Results

The proposed RVTAF depth estimation network is implemented by using Python 3.6 with Pytorch [32] 1.10.2. For hardware systems, we used a personal computer with Intel Core i7-7700K CPU (Santa Clara, CA, USA) and NVIDIA GeForce RTX 3070Ti 8G GPU (Santa Clara, CA, USA). To validate the effectiveness of our approach, we present several experimental results on challenging benchmarks that encompass diverse settings. Specifically, we provide experimental results on two famous benchmarks, which encompass both indoor and outdoor environments.

The NYU Depth V2 dataset [33] consists of 120 K image-depth pairs obtained from video sequences captured using a Microsoft Kinect (Redmond, WA, USA). The images have a size of 480 × 640 and are collected from 464 indoor scenes. For training our network, we utilize approximately 50 K training pairs obtained from random crops of size 416 × 544. We evaluate the performance of our approach on 654 testing pairs at full resolution. The depth maps have an upper bound of 10 m. The two selected image-depth pairs in the NYU Depth V2 dataset are shown in Figure 16.

Speaking of an outdoor scene, the KITTI dataset [34] is widely recognized in the field of depth estimation. The KITTI dataset comprises 61 scenes from various categories such as “city”, “residential”, “road”, and “campus”. To ensure fair comparisons with existing methods, we adopt the split proposed by Eigen et al. [18] for training and testing. Therefore, we evaluate our approach on a subset of 652 images across 29 scenes, while the remaining 32 scenes consisting of 23,488 images are used for training purposes. The RGB images have a resolution of approximately 376 × 1241, whereas the corresponding depth maps exhibit low density and contain numerous missing data points. Therefore, we calculate the loss function only for those of the depth map that have valid values. Two selected image–depth pairs in the KITTI dataset are shown in Figure 17. The images are uniformly cropped to a fixed size of 352 × 1216 at a specific position. Afterwards, we train our network using a random crop of size 352 × 704. During evaluation, we utilize the full resolution with the size of 352 × 1216.

For performance evaluation, in this paper, we use three inlier metrics and root mean square error and absolute related error matrices, where are used by previous work:

The inlier matrices, denoting by δ_**1**_, δ_**2**_, and δ_**3**_, are defined as follows:(26)$$\delta =\frac{1}{\left|T\right|}\left|\{{\hat{d}}_{i}\in T\left|\max(d_{i}/{\hat{d}}_{i},{\hat{d}}_{i}/d_{i})<{\left(1.25\right)}^{t}\}\right.\right|,\times \;100\%$$

The root mean square error (RMSE) is defined as:(27)$$\mathrm{RMSE}=\sqrt{\frac{1}{\left|T\right|}\sum_{{\hat{d}}_{i}\in T}{\left\|{\hat{d}}_{i}-d_{i}\right\|}^{2}}$$

The absolute relative error (AbsRel) is defined as:(28)$$\mathrm{AbsRel}=\frac{1}{\left|T\right|}\sum_{{\hat{d}}_{i}\in T}\left|{\hat{d}}_{i}-d_{i}\right|/d_{i}$$
where *T* denotes a collection of pixels that the ground truth values are available in and $d_{i}$ and ${\hat{d}}_{i}$ represent the ground truth depth and predicted depth of the *i*th pixel, respectively.

To prevent overfitting during network training, we employ several data augmentation techniques. For both the KITTI dataset and the NYU Depth v2 dataset, we utilize random cropping. For the NYU Depth v2 dataset, we crop the images to a size of 416 × 544 during training and perform inference with the full-size images, which are 480 × 640. For the KITTI dataset, we crop the images to a size of 352 × 704 and perform inference with a size of 352 × 1216. Additionally, each image has a 50% chance of being horizontally flipped. We also apply random adjustments to the brightness, saturation, and hue of each image. These data augmentation methods introduce variability into the training set, effectively reducing the risk of overfitting.

For training, we utilize the Adam optimizer [35] with cosine decay. We adopt the one-cycle learning rate policy. The learning rate increases by applying linear warm-up from 1 × 10^−5^ to 1 × 10^−4^ for the first 10% of iterations followed by cosine decay to 3 × 10^−5^. The total number of epochs is set to 150, with a batch size of 6, except for the ablation study, which is trained for approximately 70 epochs.

### 4.1. CNN-ViT Encoder with Various ViT Configurations

First, we conducted intensive experiments to determine a better positioning of vision transformers (ViTs) combined into the CNN encoder shown in Figure 9; the most reasonable configurations are listed in Table 1. Hereafter for all tables, it is noted that the bold results mean the best achieved ones in each metric. To reduce the computation, we only test the patterns with more ViTs for the deeper-level features, which could achieve better results and less computation. Each configuration is denoted by five digits, *n*_1_ *n*_2_ *n*_3_ *n*_4_ *n*_5_, which represent the specific numbers of ViTs used in Positions 1, 2, 3, 4, 5, respectively, as illustrated in Figure 9. For the “00000” case, it is noted that no ViT modules are used; thus, the computation becomes the smallest. For “00131”, it is indicated that there are no ViTs in Positions 1 and 2, one ViT in Position 3, three ViTs in Position 4, and one ViT in Position 5. The simulation results show that, of the position indices, “01121” has the best estimation performance. However, we prefer to choose the position index “00023”, which achieves a quality near that of “01121” and has lower computation complexity. Thus, as shown in Figure 5, the proposed CNN-ViT encoder is used with the two residual ViTs in Position 4 and three ViTs in Position 5 to complete the RVTAF autoencoder for depth estimation.

### 4.2. Adaptive Fusion Decoder with Various Fusion Modules

In the previous section, we provided a detailed introduction of the baseline method SFF [16] and three proposed fusion modules. Now, we conduct the comparative analyses of these fusion modules as the subsequent ablation study. To ensure a fair comparison of their performance, we utilized the same model architecture for all four fusion modules, only replacing the specific fusion component. The results in Table 2 clearly demonstrate that adaptive fusion module (AFM), which concatenates the skip connection feature and the decoded feature and generates attention maps through two separate branches, consistently outperforming the other fusion modules across all evaluation metrics. As expected, the AFM with concatenated data and separate branches requires a higher number of parameters. The AFM decoder is designed in the RVTAF autoencoder for depth estimation.

### 4.3. Comparisons on NYU Depth V2 Dataset

In this experiment, we utilized the NYU Depth V2 test set, specifically 654 samples, to evaluate the performance of the three models. Table 3 clearly demonstrates that our proposed RVTAF depth estimation network outperforms the other two methods across all evaluation metrics. Figure 18 shows two test images and their ground truth depth maps. These two images contain particular glass windows and long-range targets. Figure 19 shows the visualization comparisons of depth estimation results comparing with the existing approaches. The proposed RVTAF depth estimation network in all figures achieves better depth results than the BTS [30] and GLPDepth [17] methods in the NYU Depth V2 dataset. The proposed RVTAF method shows the δ_1_ accuracy metric with an improvement of about 28% and the RMSE with a reduction of about 27% compared to the GLPDepth method.

### 4.4. Comparisons on KITTI Dataset

In this experiment, we utilized the KITTI dataset Eigen split, which contains 652 testing images. In Table 4, it is evident that our proposed RVTAF network surpasses GLPDepth in all metrics, particularly in the **δ_1_** metric, where the proposed RVTAF network demonstrates similar performance, with a slight edge over the BTS.

Figure 20 shows visualization comparisons of the depth estimation results achieved by the proposed and the existing approaches in the KITTI dataset. The proposed RVTAF network in the figures also exhibits better depth results than the BTS and GLPDepth methods.

## 5. Conclusions

In this paper, we proposed a residual vision transformer and adaptive fusion (RVTAF) depth estimation network that is based on an autoencoder with skip connection architecture. In the proposed encoder, we suggest residual configurations of vision transformers (ViTS) to the CNN-based feature extractor achieve better performance. As for the proposed decoder, we introduce the adaptive fusion module (AFM) to effectively fuse the skip connected features from the encoder with the decoded feature, where the AFM generates two separate attention maps, allowing each feature to concentrate on specific spatial information. Additionally, we enhanced the decoder by incorporating a deep ASPP module to expand the effective receptive field of deep features. Ultimately, the proposed RVTAF depth estimation network is capable of accurately predicting depth maps from a single image. We conducted ablation studies to determine the best ViT configuration that uses less parameters and maintains the best performance for depth estimation and to evaluate the effectiveness of the proposed AFM. Subsequently, we compared our final network with the existing methods for both indoor scenes on the NYU Depth V2 dataset and outdoor scenes on the KITTI dataset Eigen split. In the case of indoor scenes, our method achieves sharper boundaries and more accurate depth values. Additionally, our network successfully captures depth information from traffic signs and vehicles in the KITTI dataset. Overall, the experimental results demonstrate that our method is competitive with the current methods. With the introduction of ViTs, the computation of the proposed method is slightly increased for depth estimation. The simplified ViT could be considered to promote real applications in the future.

## Figures and Tables

**Figure 1 sensors-25-00080-f001:**
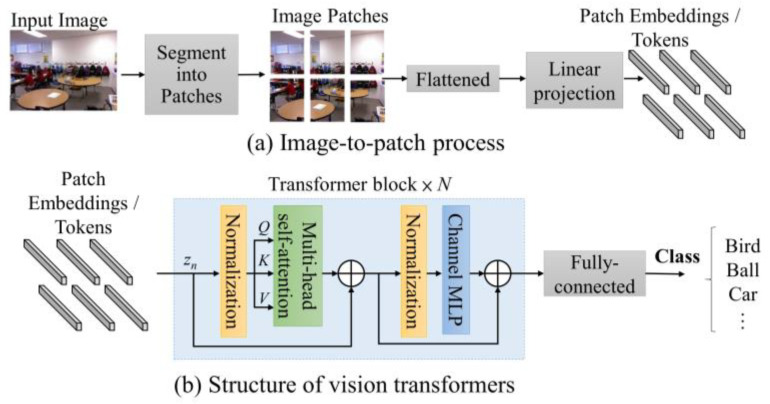
The structure of the vision transformer.

**Figure 2 sensors-25-00080-f002:**
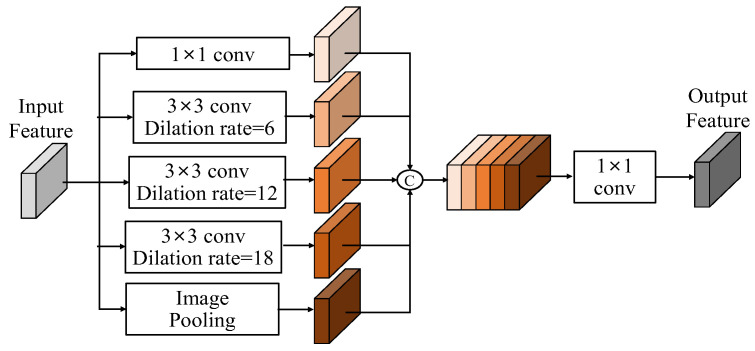
Architecture of Atrous spatial pyramid pooling.

**Figure 3 sensors-25-00080-f003:**
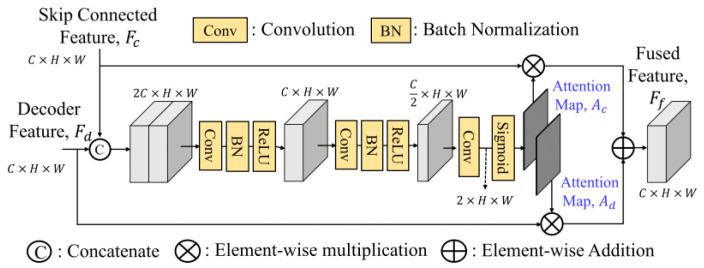
Structure of selective feature fusion (SFF) module.

**Figure 4 sensors-25-00080-f004:**
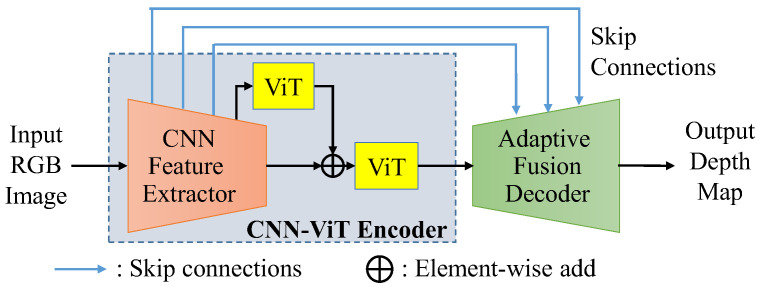
Basic framework of the proposed RVTAF depth estimation network.

**Figure 5 sensors-25-00080-f005:**
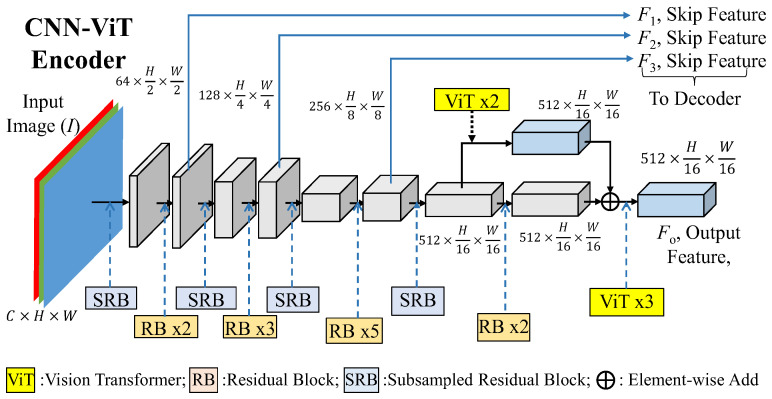
Detailed structure of the final CNN-ViT encoder.

**Figure 6 sensors-25-00080-f006:**
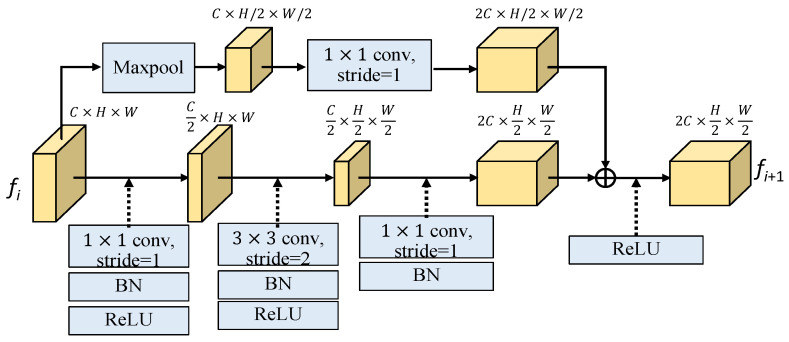
The structure of the subsampled residual block (SRB).

**Figure 7 sensors-25-00080-f007:**
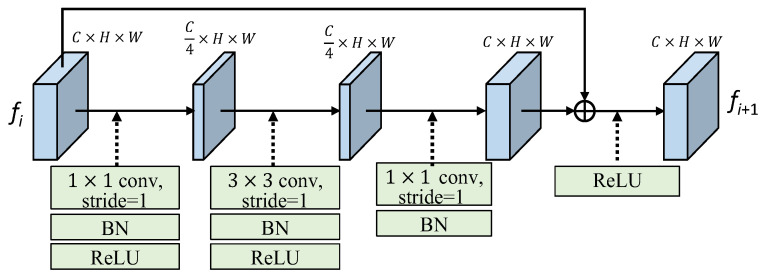
The structure of the residual block (RB).

**Figure 8 sensors-25-00080-f008:**
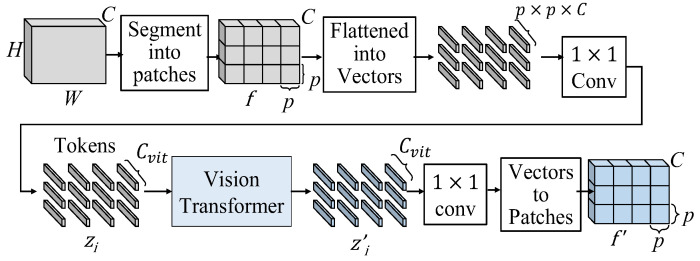
Flow chart of the realized vision transformer.

**Figure 9 sensors-25-00080-f009:**
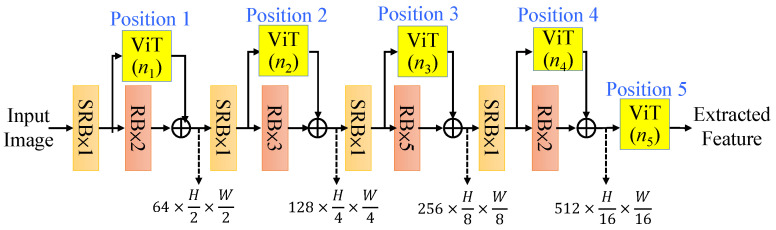
The depicted positions for inserting the residual vision transformers.

**Figure 10 sensors-25-00080-f010:**
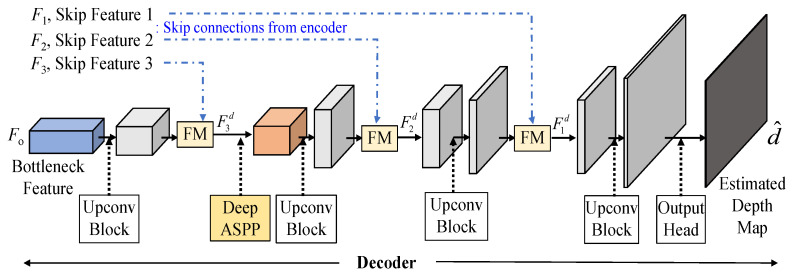
The structure of the proposed adaptive fusion decoder in the proposed RVTAF depth estimation network.

**Figure 11 sensors-25-00080-f011:**
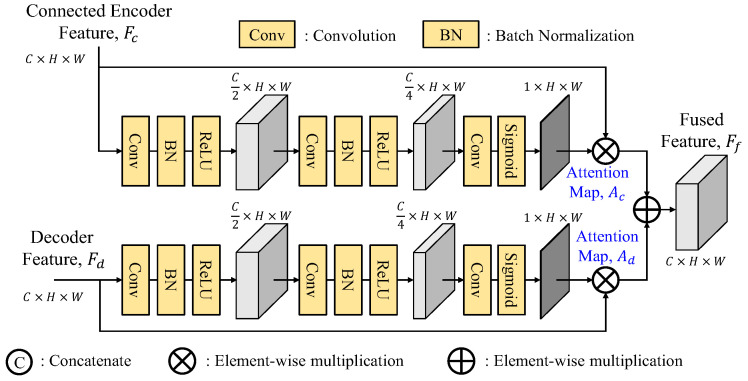
Structure of separate enhancement addition fusion module (SEAFM).

**Figure 12 sensors-25-00080-f012:**
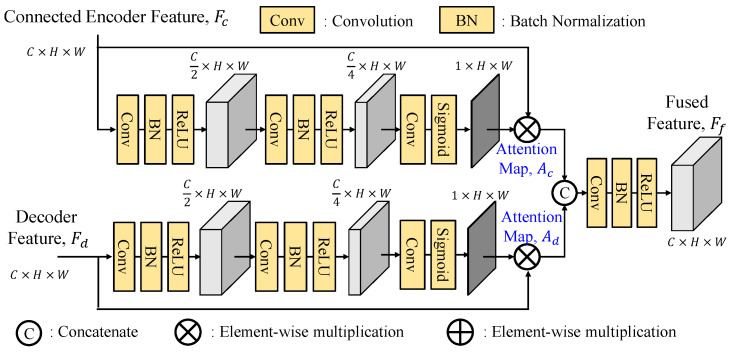
Architecture of separate enhancement concatenation fusion module (SECFM).

**Figure 13 sensors-25-00080-f013:**
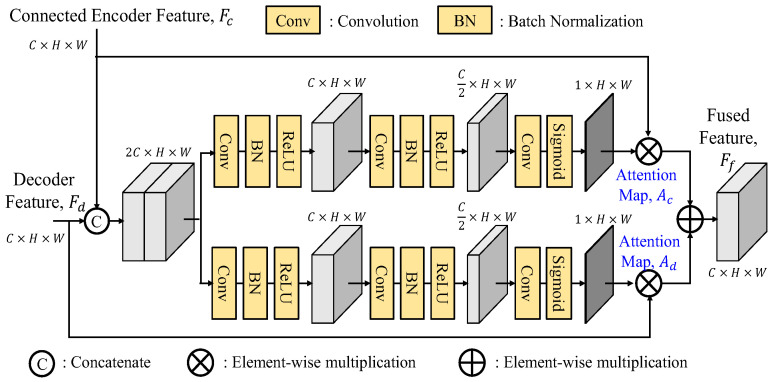
Architecture of adaptive fusion module (AFM).

**Figure 14 sensors-25-00080-f014:**
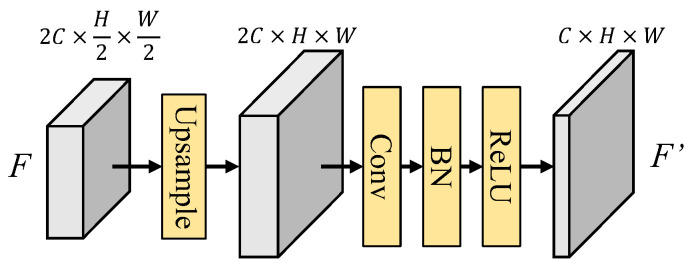
Architecture of up-convolution block.

**Figure 16 sensors-25-00080-f016:**
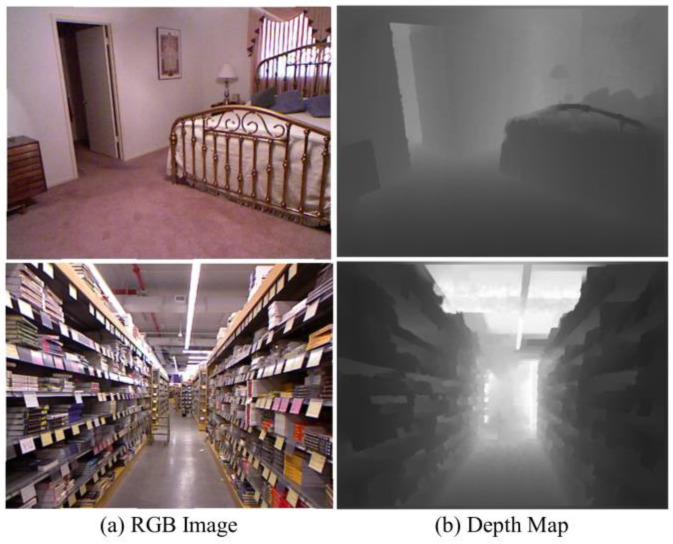
Two selected RGB color images and their corresponding depth maps in the NYU Depth V2 dataset.

**Figure 17 sensors-25-00080-f017:**
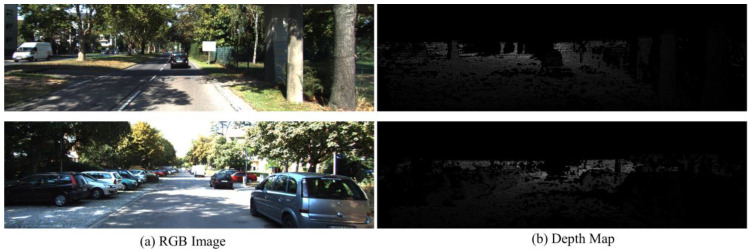
Two selected RGB color images and their corresponding depth maps in the KITTI dataset.

**Figure 18 sensors-25-00080-f018:**
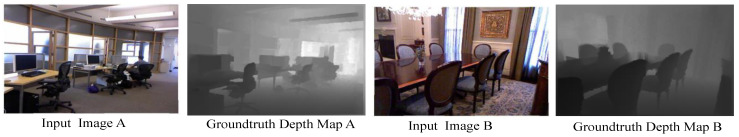
Two selected images and their corresponding ground truth depth maps on the NYU Depth V2 dataset.

**Figure 19 sensors-25-00080-f019:**
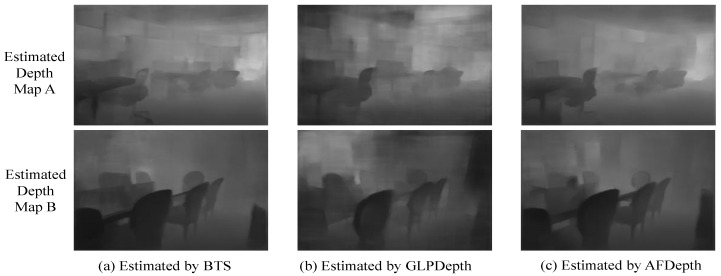
Visualizations of depth estimation results obtained with the proposed RVTAF network and the existing approaches.

**Figure 20 sensors-25-00080-f020:**
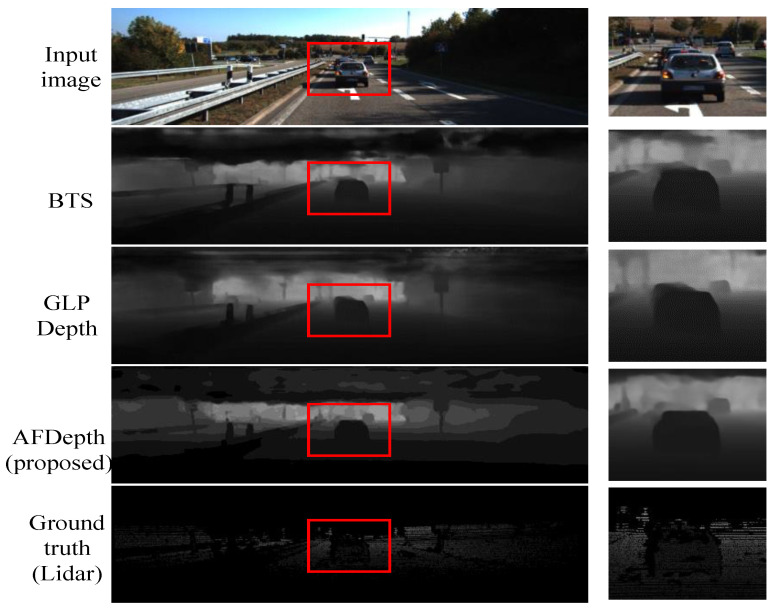
Visualization comparisons of the depth estimation achieved by the proposed RVTAF depth estimation network and the existing approaches on the KITTI dataset.

**Table 1 sensors-25-00080-t001:** Experiments of arrangements of ViTs evaluated on NYU V2 dataset Test 1449.

ViT Positions *n*_1_ *n*_2_ *n*_3_ *n*_4_ *n*_5_	Flops (G)	Params (MB)	δ_1_ ↑	δ_2_ ↑	δ_3_ ↑	RMSE ↓	AbsRel ↓
00000 (no ViTs)	**6.367**	**1.696**	0.622	0.881	0.966	0.453	0.225
00005	13.229	64.815	0.875	0.967	0.991	0.371	0.105
00014	13.522	89.828	0.879	0.968	0.991	0.366	0.106
00023	13.522	89.828	0.880	**0.971**	**0.992**	0.360	0.101
00032	13.522	89.828	0.881	0.969	0.991	0.360	0.102
00041	13.522	89.828	0.878	0.969	**0.992**	0.365	0.102
00113	14.357	102.33	0.879	0.968	0.991	0.365	0.106
00122	14.357	102.33	0.881	0.968	0.990	0.361	0.105
00131	14.357	102.33	0.876	0.968	0.990	0.370	0.109
00212	14.603	102.33	0.878	0.969	0.991	0.363	0.101
00221	14.603	102.33	0.880	0.968	0.991	0.362	0.103
00311	14.849	102.33	0.879	0.970	**0.992**	0.363	0.104
01112	16.797	108.62	0.878	0.968	0.991	0.361	0.103
01121	16.797	108.62	**0.882**	0.970	**0.992**	**0.357**	**0.100**
01211	17.043	108.62	0.874	0.967	0.990	0.364	0.104
02111	18.030	108.62	0.880	0.968	0.991	0.358	0.105
11111	24.317	112.31	0.878	0.967	0.991	0.364	0.103

**Table 2 sensors-25-00080-t002:** Experimental results with variations in fusion modules on NYU V2 testset 654.

Fusion Modules	Params (MB)	δ_1_ ↑	δ_2_ ↑	δ_3_ ↑	RMSE ↓	AbsRel ↓
SFF (baseline)	1.665	0.696	0.907	0.971	0.651	0.206
SEAFM	**0.836**	0.718	0.919	0.975	0.615	0.192
SECFM	2.159	0.717	0.917	0.973	0.626	0.195
AFM	3.320	**0.747**	**0.930**	**0.978**	**0.589**	**0.181**

**Table 3 sensors-25-00080-t003:** Comparison with existing approaches on NYU Depth V2 test set 654.

Network	δ_1_ ↑	δ_2_ ↑	δ_3_ ↑	RMSE ↓	AbsRel ↓
BTS [30]	0.762	0.940	**0.984**	0.565	0.167
GLPDepth [17]	0.605	0.872	0.962	0.769	0.235
RVTAF Net	**0.773**	**0.942**	**0.984**	**0.560**	**0.162**

**Table 4 sensors-25-00080-t004:** Comparisons the proposed RVTAF depth network and the existing approaches on KITTI dataset Eigen split.

Network	δ_1_ ↑	δ_2_ ↑	δ_3_ ↑	RMSE ↓	AbsRel ↓
BTS [30]	0.899	**0.979**	0.994	3.734	**0.093**
GLPDepth [17]	0.876	0.970	0.992	3.776	0.108
RVTAF Net	**0.902**	**0.979**	**0.995**	**3.634**	0.094

## Data Availability

The authors use KITTI dataset in [28] and the NYU Depth v2 dataset in [24] for all simulations.

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
