# Peer review of "Residual Vision Transformer and Adaptive Fusion Autoencoders for Monocular Depth Estimation"

_sensors, 2024, doi:10.3390/s25010080_

Round 1
Reviewer 1 Report
Comments and Suggestions for Authors
In this paper, the authors proposed a residual vision transformer and adaptive fusion (RVTAF) depth estimation network that is based on an autoencoder with skip connection architecture. In this encoder, an enhanced residual configuration based on a CNN-based feature extractor is proposed. For the proposed decoder, an Adaptive Fusion Module (AFM) was introduced to effectively combine the skip-connection features from the encoder and the decoded features. The effectiveness of the method is verified by experiments. The method is with novelty. However, the following points should be addressed before the consideration of acceptance.
1. The proposed method is only compared with two methods. However, there are many deep learning-based depth estimation methods. For example, the following are related to the proposed methods as they are also using CNN and Transformer in their network models for depth estimation. So, the proposed method should be compared with more methods related to their model to show their method’s advantages.(1)Zezheng Zhang, Ryan K.Y. Chan, Kenneth K.Y. Wong,GlocalFuse-Depth: Fusing transformers and CNNs for all-day self-supervised monocular depth estimation, Neurocomputing, 569, 2024,127122.
(2) X. Zhu, Z. Han, Z. Zhang, L. Song, H. Wang and Q. Guo, “PCTNet: depth estimation from single structured light image with a parallel CNN-transformer network,” Measurement Science and Technology, 2023.
2. In the introduction, there is not enough references of other researchers in the field of depth estimation.
3. There is a formatting error in the formulas of this article; please make the necessary corrections.
4. The shortcomings of the proposed method should be mentioned clearly such as in conclusion part.
Author Response
Reviewer 1:
Comments and Suggestions for Authors
In this paper, the authors proposed a residual vision transformer and adaptive fusion (RVTAF) depth estimation network that is based on an autoencoder with skip connection architecture. In this encoder, an enhanced residual configuration based on a CNN-based feature extractor is proposed. For the proposed decoder, an Adaptive Fusion Module (AFM) was introduced to effectively combine the skip-connection features from the encoder and the decoded features. The effectiveness of the method is verified by experiments. The method is with novelty. However, the following points should be addressed before the consideration of acceptance.
- The proposed method is only compared with two methods. However, there are many deep learning-based depth estimation methods. For example, the following are related to the proposed methods as they are also using CNN and Transformer in their network models for depth estimation. So, the proposed method should be compared with more methods related to their model to show their method’s advantages.
(1) Zezheng Zhang, Ryan K.Y. Chan, Kenneth K.Y. Wong, GlocalFuse-Depth: Fusing transformers and CNNs for all-day self-supervised monocular depth estimation, Neurocomputing, 569, 2024,127122.
(2) X. Zhu, Z. Han, Z. Zhang, L. Song, H. Wang and Q. Guo, “PCTNet: depth estimation from single structured light image with a parallel CNN-transformer network,” Measurement Science and Technology, 2023.
Response: Yes, In the revised manuscript, we added the reviews of your mentioned two methods in the introduction section. These two methods are good papers, however, these two papers need special datasets. So, we added the statement as “To use local and details and global dependency, the GlobalFuse-Depth uses CNN and Transformer branches to capture both features and fuse the night and daytime to-gether to achieve better estimation on Oxford RobotCar dataset [31]. The network needs a pair of night and daytime images, which can be obtained with a pretrained CycleGAN. The PCTNet also adopts the architecture of CNN and Transformer branches and uses bidirectional feature and cross-feature multi-scale fusion modules to obtain better results for single structured light image depth estimation. Glob-alFuse-Depth needs a paired dataset and a pretrained model while PCTNet uses self-supervised tests with structured light databases.” Our method is different in uses of dataset from these two methods. Besides, these two papers do not provide sources codes. The fair comparisons in a short time become impossible in this moment.
- In the introduction, there is not enough references of other researchers in the field of depth estimation.
Response: Yes, based on your suggestions, beside the above statement, we have added more related references and cited them more in Introduction section.
- There is a formatting error in the formulas of this article; please make the necessary corrections.
Response: Yes, in revised manuscript, we add more formulations to explain the functions of the proposed network modules and add three more equations to state the evaluation matrices and we have tried to correct the formatting error.
- The shortcomings of the proposed method should be mentioned clearly such as in conclusion part.
Response: Comparing to the recent work, BTS and GLPDepth, the shortcoming of the proposed method is the computation introduced by ViT module, we have stated it in conclusion part. We added the statement, “With the introduction of ViTs, the computation of the proposed method is slightly increased. The simplified ViT could be considered in the future.” in the end of conclusion.

Reviewer 2 Report
Comments and Suggestions for Authors
Title: RVTAF: Residual Vision Transformer and Adaptive Fusion Autoencoders for Monocular Depth Estimation
===============================================
Decision: Reject
=============
1- The manuscript presents an integration of residual vision transformers and adaptive fusion modules into an autoencoder architecture for monocular depth estimation. While combining these well-established deep learning techniques is an interesting approach, the paper does not adequately demonstrate significant novelty or contribution to the field. The proposed RVTAF network appears to be an incremental improvement rather than a substantial advancement over existing methods.
2- The clarity and rigor of the manuscript are areas of concern. Key sections, including the introduction and methodology, lack clear articulation of the proposed innovations. The explanations of technical concepts such as the adaptive fusion module and the training loss function are verbose yet imprecise, making it challenging for readers to grasp the core ideas. The structure of the paper is somewhat disorganized, with frequent digressions into background material that overshadow the authors' contributions.
3- In terms of experimental validation, the results provided are insufficient to substantiate the claimed superiority of the RVTAF network. Although there are marginal improvements over baseline methods on certain metrics, these gains are minimal and may not justify the increased complexity introduced by the proposed architecture. The evaluation is limited to standard benchmarks like the NYU Depth V2 and KITTI datasets, without exploring additional datasets or scenarios to demonstrate generalizability and robustness.
4- The manuscript also lacks a thorough comparison with recent state-of-the-art methods in monocular depth estimation. Essential related works are either omitted or only briefly mentioned, without proper critical analysis. This omission makes it difficult to contextualize the contributions within the existing body of research. Moreover, the paper relies heavily on citations of prior work in the methodology section, often at the expense of in-depth explanations of the proposed innovations.
5- Technical limitations are evident in the computational complexity of the RVTAF network. The model has a significantly higher number of parameters and computational demands compared to simpler, more efficient models. This raises practical concerns regarding its deployment in resource-constrained environments. Additionally, the training process lacks adequate justification for the choice of loss functions and optimization strategies, which are crucial for understanding the network's performance.
6- The presentation quality of the manuscript requires substantial improvement. Figures and tables are poorly labeled and lack detailed explanations, making it difficult to interpret the data and experimental results effectively. For example, the presentation of results in Tables 1, 2, and 5 is cluttered, hindering the identification of key findings. The manuscript also contains grammatical errors and uses language that does not meet the professional standards expected in scholarly publications.
while the integration of residual vision transformers and adaptive fusion modules into an autoencoder for monocular depth estimation is a promising idea, the current manuscript falls short in demonstrating significant contributions to the field. The lack of clear novelty, insufficient experimental validation, inadequate comparison with related work, and issues with clarity and presentation are substantial barriers to publication.
============================
Recommendations for the Authors:
============================
To improve the manuscript, it is recommended that the authors refocus their work to clearly highlight the specific contributions of the RVTAF network and how it advances the state of the art in monocular depth estimation. They should perform a more extensive experimental evaluation, including additional benchmarks and comprehensive ablation studies, to validate the robustness and applicability of the proposed method. Addressing computational inefficiencies and providing practical guidelines for implementation would strengthen the practical relevance of the work. Enhancing the clarity and quality of figures, tables, and the overall presentation is also essential. Finally, a thorough proofreading is necessary to correct grammatical errors and improve the professional tone of the manuscript.
Given these concerns, I cannot recommend this manuscript for publication in its current state. A significant revision addressing the issues mentioned is required before it can be reconsidered.
Comments on the Quality of English LanguageA thorough proofreading is necessary to correct grammatical errors and improve the professional tone of the manuscript.
Author Response
Comments and Suggestions for Authors
Title: RVTAF: Residual Vision Transformer and Adaptive Fusion Autoencoders for Monocular Depth Estimation
===============================================
Decision: Reject
=============
1- The manuscript presents an integration of residual vision transformers and adaptive fusion modules into an autoencoder architecture for monocular depth estimation. While combining these well-established deep learning techniques is an interesting approach, the paper does not adequately demonstrate significant novelty or contribution to the field. The proposed RVTAF network appears to be an incremental improvement rather than a substantial advancement over existing methods.
Response: Yes, thanks for your criticized comments. We have completely rewritten the whole manuscripts to emphasize the contribution of the paper. Up to now, we first proposed the residual ViTs, which could be inserted in the CNN to improve the depth estimation. Simulation results demonstrate the improvement of the proposed method for monocular depth estimation.
2- The clarity and rigor of the manuscript are areas of concern. Key sections, including the introduction and methodology, lack clear articulation of the proposed innovations. The explanations of technical concepts such as the adaptive fusion module and the training loss function are verbose yet imprecise, making it challenging for readers to grasp the core ideas. The structure of the paper is somewhat disorganized, with frequent digressions into background material that overshadow the authors' contributions.
Response: Thanks for your valuable comments, In the revised manuscript, we try our best to clearly explain the proposed methods All the weakness points raised by you have been tried to correct and revise as possible. We believe that the revised manuscript now becomes readable and technical sounded.
3- In terms of experimental validation, the results provided are insufficient to substantiate the claimed superiority of the RVTAF network. Although there are marginal improvements over baseline methods on certain metrics, these gains are minimal and may not justify the increased complexity introduced by the proposed architecture. The evaluation is limited to standard benchmarks like the NYU Depth V2 and KITTI datasets, without exploring additional datasets or scenarios to demonstrate generalizability and robustness.
Response: Thanks for your kind comments. We added the performance improvement in metrics in the abstract of the revised paper. In the end of abstract, we added a statement “which increases the first accuracy rate about 28% and reduces the root mean square error about 27% than an existing method in NYU dataset.” to support the contribution of our paper. The NYU Depth V2 and KITTI datasets, which are the most common benchmarks. By using supervised learning approaches, we believe that the availabilities and the generalizability of evaluations are still reasonable for academic institutes.
4- The manuscript also lacks a thorough comparison with recent state-of-the-art methods in monocular depth estimation. Essential related works are either omitted or only briefly mentioned, without proper critical analysis. This omission makes it difficult to contextualize the contributions within the existing body of research. Moreover, the paper relies heavily on citations of prior work in the methodology section, often at the expense of in-depth explanations of the proposed innovations.
Response: Thanks for your suggestion. The major contributions of the papers are the residual configuration of ViTs, the adaptive fusion module and the use of the deepASPP. Our methods were inspired by BTS [26] and GLPDepth [16], whose performances were claimed as the state-of-the-art in the papers. These methods outperform many existed methods. Without comparing those existed methods, the comparisons to these two methods are reasonable.
5- Technical limitations are evident in the computational complexity of the RVTAF network. The model has a significantly higher number of parameters and computational demands compared to simpler, more efficient models. This raises practical concerns regarding its deployment in resource-constrained environments. Additionally, the training process lacks adequate justification for the choice of loss functions and optimization strategies, which are crucial for understanding the network's performance.
Response: Thanks for your comment. The proposed methods are focused on the use of least computation modules to improve the estimation performance. Table The major contributions of the papers are the residual configuration of ViTs, the adaptive fusion module and the use of the deepASPP. These contributions only introduce a few computation complexities. As to the training process, we have revised it.
6- The presentation quality of the manuscript requires substantial improvement. Figures and tables are poorly labeled and lack detailed explanations, making it difficult to interpret the data and experimental results effectively. For example, the presentation of results in Tables 1, 2, and 5 is cluttered, hindering the identification of key findings. The manuscript also contains grammatical errors and uses language that does not meet the professional standards expected in scholarly publications.
Response: Thanks for your comment. In the revised manuscript, in figures, functionalities and evaluation metrics, we have added many more notations and formulations to explain them. As to grammatical errors, the revisions of the paper are assisted by an English tutor with engineering expertise have been greatly reduces.
while the integration of residual vision transformers and adaptive fusion modules into an autoencoder for monocular depth estimation is a promising idea, the current manuscript falls short in demonstrating significant contributions to the field. The lack of clear novelty, insufficient experimental validation, inadequate comparison with related work, and issues with clarity and presentation are substantial barriers to publication.
Response: Thanks for your comments. In the revised manuscript, after intensive efforts, we have explained the details and properly added proper statements for further improvement of readabilities.
============================
Recommendations for the Authors:
============================
To improve the manuscript, it is recommended that the authors refocus their work to clearly highlight the specific contributions of the RVTAF network and how it advances the state of the art in monocular depth estimation. They should perform a more extensive experimental evaluation, including additional benchmarks and comprehensive ablation studies, to validate the robustness and applicability of the proposed method. Addressing computational inefficiencies and providing practical guidelines for implementation would strengthen the practical relevance of the work. Enhancing the clarity and quality of figures, tables, and the overall presentation is also essential. Finally, a thorough proofreading is necessary to correct grammatical errors and improve the professional tone of the manuscript.
Response: Thanks for your kind comments. In the revised manuscript, after intensive efforts, we have explained the details and properly added proper statements for further improvement of readabilities.
Given these concerns, I cannot recommend this manuscript for publication in its current state. A significant revision addressing the issues mentioned is required before it can be reconsidered.
Response: Thanks for your criticized comments. We have completely rewritten the whole manuscripts to emphasize the contribution of the paper. Up to now, we first proposed the residual ViTs, which could be inserted in the CNN to improve the depth estimation. Simulation results demonstrate the improvement of the proposed method for monocular depth estimation.
Comments on the Quality of English Language
A thorough proofreading is necessary to correct grammatical errors and improve the professional tone of the manuscript.
Response: Thanks for your comment. In the revised manuscript, in figures, functionalities and evaluation metrics, we have added many more notations and formulations to explain them. As to grammatical errors, the revisions of the paper are assisted by an English tutor with engineering expertise have been greatly reduces.

Reviewer 3 Report
Comments and Suggestions for Authors
Overall Decision: Major revision
This manuscript presents an innovative approach to monocular depth estimation through residual vision converters and adaptive fusion autoencoders. While the research demonstrates promising potential, several critical revisions are necessary to elevate the manuscript to publication standards.
(1) The integration of residual vision transformers and adaptive fusion modules presents a novel approach to monocular depth estimation, with significant applications in autonomous driving and virtual reality. However, the manuscript should more thoroughly address the inherent limitations of single-view depth estimation, particularly regarding complex occlusions and low-texture regions. A more rigorous discussion of these challenges would strengthen the paper's theoretical foundation.
(2) The rationale behind selecting Cosine Decay over alternative strategies (line 353) requires clarification.
(3) The experimental results section lacks specific quantitative indicators and comparative data, diminishing its persuasiveness. Include concrete performance metrics and benchmarking data.
(4) The introduction requires a broader perspective and CV application on the research topic, supported by authoritative references, For instance, Deep Neural Remote Sensing and Sentinel-2 Satellite Image Processing of Kirkuk City, Iraq for Sustainable Prospective. Journal of Optics and Photonics Research. 3D vision technologies for a self-developed structural external crack damage recognition robot; Automation in Construction.
(5) The study's novelty needs stronger emphasis through clear differentiation from existing work in the field.
(6) While the proposed detection techniques are sound, the work's limitations require more thorough justification and analysis.
(7) Include transition statements and a concise presentation of the research framework to improve flow and readability.
(8) Section 2.1 should elaborate on ViT's specific advantages over traditional CNNs in depth estimation tasks, rather than focusing solely on ViT's general structure.
(9) Add a theoretical foundation explaining how ViT modules effectively complement CNN's local feature extraction in depth estimation.
(10) Standardize the presentation of computational tools and software resources across all methodology sections.
(11) Enhance the clarity of figures 6 and 7 by using distinct colors for feature flow and data transformation arrows.
(12) Provide theoretical justification for the performance improvements achieved through "more ViTs for deep features" in section 4.1.
(13) Expand Figure 20 to include diverse sample comparisons, particularly focusing on complex textures and long-range targets.
(14) The conclusion should address future research directions and practical applications.
(15) Incorporate a more comprehensive analysis of the study's limitations in this review.
Author Response
Comments and Suggestions for Authors
Overall Decision: Major revision
This manuscript presents an innovative approach to monocular depth estimation through residual vision converters and adaptive fusion autoencoders. While the research demonstrates promising potential, several critical revisions are necessary to elevate the manuscript to publication standards.
(1) The integration of residual vision transformers and adaptive fusion modules presents a novel approach to monocular depth estimation, with significant applications in autonomous driving and virtual reality. However, the manuscript should more thoroughly address the inherent limitations of single-view depth estimation, particularly regarding complex occlusions and low-texture regions. A more rigorous discussion of these challenges would strengthen the paper's theoretical foundation.
Response: Based 3D geometry, to find a distance of the object, we need at least two cameras to obtain the results. Thus, the monocular depth estimation is treated as an ill-posed problem. When humans try to understand the spatial relationship of the objects in a view-single (i.e., one-eye) image, we need to explore the local and global information. Heavily depending on the image to depth mapping functions, the monocular depth estimation methods need proper image and it groundtruth pairs. That is why we need to explore more global feature.
(2) The rationale behind selecting Cosine Decay over alternative strategies (line 353) requires clarification.
Response: For training, we utilize the Adam optimizer [29] with cosine decay. We adopt the one-cycle learning rate policy. The learning rate increases by applying linear warm-up from 1e-5 to 1e-4 for the first 10% of iterations followed by cosine decay to 3e-5. The total number of epochs is set to 150, with a batch size 6, except for the ablation study, which is trained for approximately 70 epochs.
(3) The experimental results section lacks specific quantitative indicators and comparative data, diminishing its persuasiveness. Include concrete performance metrics and benchmarking data.
Response: Yes, for experimental results, we added the definitions of evaluation metrics by adding 3 formulations for quality comparison. The equations (26), (27) and (28) define the inlier matrices, denoting d1, d2, and d3,RMSE and AbsRel, respectively.
(4) The introduction requires a broader perspective and CV application on the research topic, supported by authoritative references, For instance, Deep Neural Remote Sensing and Sentinel-2 Satellite Image Processing of Kirkuk City, Iraq for Sustainable Prospective. Journal of Optics and Photonics Research. 3D vision technologies for a self-developed structural external crack damage recognition robot; Automation in Construction.
Response: Yes, the precision estimation can help to achieve more CV applications.
(5) The study's novelty needs stronger emphasis through clear differentiation from existing work in the field.
Response: Yes, in end of the conclusion, we added the statement, “The simplified ViT could be considered to promote real applications in the future.”.
(6) While the proposed detection techniques are sound, the work's limitations require more thorough justification and analysis.
Response: Yes, in end of the conclusion, we added the statement, “The simplified ViT could be considered to promote real applications in the future.”.
(7) Include transition statements and a concise presentation of the research framework to improve flow and readability.
Response: Yes, by adding mathematical expressions, we used formulations to concise presentation of research framework. We believe that these formulations can help the readers to understand all steps and details of the proposed method.
(8) Section 2.1 should elaborate on ViT's specific advantages over traditional CNNs in depth estimation tasks, rather than focusing solely on ViT's general structure.
Response: Yes, in Section 2.1, we emphasize the global feature can be further improved by ViTs, while the detailed computation process in formulations of ViT process has been present in Section 3.1.2. We believe that the readers can fast learn the ViT advantages and details.
(9) Add a theoretical foundation explaining how ViT modules effectively complement CNN's local feature extraction in depth estimation.
Response: Yes, in Section 3.1, we completely rewrote the presentation of CNN-ViT encoder by adding feature notations and their detailed formulation. Thus, the readers can easily learn the theoretical foundation of residual ViT. The detailed computation process in formulations of ViT process has been present in Section 3.1.2.
(10) Standardize the presentation of computational tools and software resources across all methodology sections.
Response: Yes, Yes, by mathematical expressions, including revised figures, we have added formulations to explain all the functions and evaluation metrics. We believe that these formulations can help to understand all methodology.
(11) Enhance the clarity of figures 6 and 7 by using distinct colors for feature flow and data transformation arrows.
Response: Yes, we have re-colored Figures 6 for SRB and 7 for RB to match up with the color blocks in Figure 5. To help the details, we also added formulations to explain the functionalities.
(12) Provide theoretical justification for the performance improvements achieved through "more ViTs for deep features" in section 4.1.
Response: In general, the deep features are with more global information. The ViTs are also focused on retrieving global information. Adding more ViTs for deep features are reasonable. However, we adopt the residual ViT designs, we thus use the intensive experiments to explore a better configuration, instead of detailed theoretical justification.
(13) Expand Figure 20 to include diverse sample comparisons, particularly focusing on complex textures and long-range targets.
Response: These two images are selected for NYU database, we add more explanation about these images with a sentence as, ”These two images contain particular glass windows and long-range targets.”
(14) The conclusion should address future research directions and practical applications.
Response: Yes, in the revised manuscript, we added the statement, “The simplified ViT could be considered to promote real applications in the future.”.
(15) Incorporate a more comprehensive analysis of the study's limitations in this review.
Response: Yes, comparing to the existing methods, the high computation of the proposed method is highlighted.

Reviewer 4 Report
Comments and Suggestions for Authors
In the paper, the authors proposed a residual vision transformer and adaptive fusion (RVTAF) depth estimation network based on an autoencoder with skip connection architecture. The obtained results are encouraging. Regarding the structure and the quality of the paper, the following observations can be made:
1. Avoid the acronym RVTAF from the title. The complete terms has already been integrated.
2. Any abbreviations (CNN-ViT and RVTAF) should be removed in the abstract section. A special paragraph should be introduced in the main body of the article. Considering the numerous abbreviations used within the manuscript, an abbreviation list is recommended. Also, the authors should provide the meaning of all abbreviations.
3. On the other hand, the authors asserted in the abstract that “Experimental results demonstrate the effective prediction of the depth map from a single-view colour image by the proposed RVTAF autoencoder.” The values of the used metrics should be introduced to highlight the efficacy of the proposed approach.
4. The authors should remove the reference loop [7-10], and details on each reference should be presented.
5. The introduction could be improved with a better state-of-the-art to include more techniques-related applications. The introduction summarizes, but without highlighting clear the gaps. Yes, the authors highlight a few issues, but it will be useful to highlight the gaps. After that, a synthesis of the solutions proposed, depending on the type of analysis, to highlight the advantages and disadvantages is necessary for readers. This synthesis can be given as a table.
6. The innovations of the paper should be identified in the form of headlines.
7. The structure of the paper should be ensured.
8. For Section 3, the introductions of the methodology should be refined with a mathematical tool well done to understand all steps of the proposed approach.
9. Paragraph 4.3 compares the proposed approach and the BTS [26] and GLPdepth [16] methods in the NYU Depth V2 dataset. But, the BTS appeared suddenly in comparison. Please explain the details of this technique in previous sections.
10. The limits of the proposed approach should be better presented and discussed.
11. The conclusions must be precise regarding the aim, key findings, contributions, and future research directions.
Author Response
Comments and Suggestions for Authors
In the paper, the authors proposed a residual vision transformer and adaptive fusion (RVTAF) depth estimation network based on an autoencoder with skip connection architecture. The obtained results are encouraging. Regarding the structure and the quality of the paper, the following observations can be made:
- Avoid the acronym RVTAF from the title. The complete terms has already been integrated.
Response: Yes, we have removed “RVTAF” from the title of the paper
- Any abbreviations (CNN-ViT and RVTAF) should be removed in the abstract section. A special paragraph should be introduced in the main body of the article. Considering the numerous abbreviations used within the manuscript, an abbreviation list is recommended. Also, the authors should provide the meaning of all abbreviations.
Response: Yes, we have removed the abbreviations, “CNN-ViT and RVTAF” from the abstract of the paper
- On the other hand, the authors asserted in the abstract that “Experimental results demonstrate the effective prediction of the depth map from a single-view colour image by the proposed RVTAF autoencoder.” The values of the used metrics should be introduced to highlight the efficacy of the proposed approach.
Response: Yes, we added the performance improvement in metrics in the abstract of the revised paper. In the end of abstract, we added “which increases the first accuracy rate about 28% and reduces the root mean square error about 27% than an existing method in NYU dataset.”, Corresponding to this statement, we also added an addition statement “particularly in the d1 accuracy metric with about 28% improvement and the RMSE with about 27% reduction,” in simulation section.
- The authors should remove the reference loop [7-10], and details on each reference should be presented.
Response: Yes, we removed the reference loop and detailed the reference one-by -one by replacing the statement as “Depth estimation by stereo matching neural networks was first initiated by comparing image patches [7]. To further improve the performance, the two-stage network with cascade residual learning [8], the pyramid network [9] and the semi-global and mutual information were proposed. The stereo matching neural networks, however, need multiple cameras for depth prediction.”
- The introduction could be improved with a better state-of-the-art to include more techniques-related applications. The introduction summarizes, but without highlighting clear the gaps. Yes, the authors highlight a few issues, but it will be useful to highlight the gaps. After that, a synthesis of the solutions proposed, depending on the type of analysis, to highlight the advantages and disadvantages is necessary for readers. This synthesis can be given as a table.
Response: Yes, we added a good review paper of depth estimation [33]. Starting from it, the introduction is reorganized to meet your suggestion. Some of the key references, which are highly related to our work, are then discussed in the Related Work section.
- The innovations of the paper should be identified in the form of headlines.
Response: Yes, the innovations of the paper are the designs of residual ViTs with CNN encoder and the adaptive fusion module and have been identified in the abstract and conclusions.
- The structure of the paper should be ensured.
Response: Yes, we took all the suggestions raised by the four Reviewers, the structure of the paper becomes more readable and smoothly.
- For Section 3, the introductions of the methodology should be refined with a mathematical tool well done to understand all steps of the proposed approach.
Response: Yes, by mathematical expressions, we have added formulations to explain all the functions and evaluation metrics. We believe that these formulations can help the readers to understand all steps and details of the proposed method.
- Paragraph 4.3 compares the proposed approach and the BTS [26] and GLPdepth [16] methods in the NYU Depth V2 dataset. But, the BTS appeared suddenly in comparison. Please explain the details of this technique in previous sections.
Response: Yes, we have added the review of BTS in the introduction after the review of ASSP in Subsection 2.2. The added review statement is “The BTS network [26] is composed of dense feature extractor, ASPP as the contextual information extractor, local planar guidance layers and their dense connection for depth estimation, where the ASPP can capture large scale variations in observation by applying sparse convolutions with various dilation rates. In that moment, the BTS network presented a supervised monocular depth estimation network to achieve state-of-the-art results.”. The BTS method uses the ASSP to improve the performance of depth estimation. However, we use deep ASSP for the further improvement.
- The limits of the proposed approach should be better presented and discussed.
Response: Yes, we have added the shortcoming and future work in the conclusion. Comparing to the existing methods, the high computation of the proposed method is highlighted.
- The conclusions must be precise regarding the aim, key findings, contributions, and future research directions.
Response: Yes, we have rewritten the conclusion to meet your suggestion. In the revised manuscript, we have stated the three key contributions, the residual configuration of ViTs, the design of adaptive fusion module, and the deep ASPP module.

Round 2
Reviewer 1 Report
Comments and Suggestions for Authors
The paper could be published, since the comments from the reviewer are suitably addressed.
Reviewer 3 Report
Comments and Suggestions for Authors
accept
Reviewer 4 Report
Comments and Suggestions for Authors
The authors transformed the initial version of the manuscript with new explanations, elaborations of details and revisions. I have no more observations.